# mPTP Proteins Regulated by Streptozotocin-Induced Diabetes Mellitus Are Effectively Involved in the Processes of Maintaining Myocardial Metabolic Adaptation

**DOI:** 10.3390/ijms21072622

**Published:** 2020-04-09

**Authors:** Natalia Andelova, Iveta Waczulikova, Ivan Talian, Matus Sykora, Miroslav Ferko

**Affiliations:** 1Centre of Experimental Medicine, Slovak Academy of Sciences, Institute for Heart Research, 84104 Bratislava, Slovakia; usrdnata@savba.sk (N.A.); usrdsyko@savba.sk (M.S.); 2Division of Biomedical Physics, Department of Nuclear Physics and Biophysics, Faculty of Mathematics, Physics and Informatics, Comenius University, 84248 Bratislava, Slovakia; waczulikova@fmph.uniba.sk; 3Department of Medical and Clinical Biophysics, Faculty of Medicine, P. J. Safarik University in Kosice, 04011 Kosice, Slovakia; ivan.talian@upjs.sk

**Keywords:** mitochondrial permeability transition pores, cardioprotection, experimental diabetes mellitus, proteomic analysis, endogenous protective processes

## Abstract

Mitochondrial permeability transition pores (mPTPs) have become an important topic in investigating the initiation and signaling pathways involved in cardioprotection. Experimental streptozotocin-induced diabetes mellitus (D) was shown to provide sufficient protection to the myocardium via compensatory mechanisms enabling mitochondria to produce energy under pathological conditions during the acute phase. The hypothesized involvement of mPTPs in these processes prompted us to use liquid chromatography and mass spectrometry-based proteomic analysis to investigate the effects of the acute-phase D condition on the structural and regulatory components of this multienzyme complex and the changes caused by compensation events. We detected ADT1, ATP5H, ATPA, and ATPB as the most abundant mPTP proteins. The between-group differences in protein abundance of the mPTP complex as a whole were significantly upregulated in the D group when compared with the control (C) group (*p* = 0.0106), but fold changes in individual protein expression levels were not significantly altered except for ATP5H, ATP5J, and KCRS. However, none of them passed the criterion of a 1.5-fold change in differential expression for biologically meaningful change. Visualization of the (dis-)similarity between the C and D groups and pairwise correlations revealed different patterns of protein interactions under the C and D conditions which may be linked to endogenous protective processes, of which beneficial effects on myocardial function were previously confirmed.

## 1. Introduction

Ischemic heart disease remains the leading cause of morbidity and mortality worldwide, therefore, mechanisms leading to cardioprotection are becoming a growing research issue. As a cardioprotective strategy for decreasing the vulnerability of the myocardium to ischemia/reperfusion (I/R) injury, ischemic preconditioning (PC) has received great attention for its potent infarct size-limiting effect in experiments using healthy animals [1,2]. There is an unequivocal consensus that the mitochondria make up the most important effector of PC protection, where most, if not all, signaling pathways converge [3,4,5]. Myocardial protection by PC is achieved by the activation of multiple protective signaling pathways hypothesized to inhibit mitochondrial permeability transition pore (mPTP) opening upon reperfusion [6,7,8,9,10]. The mPTP is a nonspecific, highly conductive channel enabling passage of molecules smaller than 1.5 kDa through the inner mitochondrial membrane (IMM). The pores are primarily formed and opened upon an increase in Ca^2+^ in the mitochondrial matrix, signaling by reactive oxygen species (ROS) or inorganic phosphates, or intracellular acidification [8,11,12,13,14].

mPTPs are unique not only due to their multifunctional nature, but also due to their complex dynamic structures. Unlike typical membrane pores, the structure and composition of mPTPs constantly change with respect to the cell’s actual requirements [12,15,16]. The exact molecular identity of mPTP components remains subject to debate and investigation among researchers [8,15].

The mPTP is probably a multiprotein complex containing individual proteins with structural or regulatory functions [6,12,17]. An in-depth analysis of these protein–protein interactions was assumed to help uncover a potential mechanism of mPTP regulation. The components falling within the current concept of the mPTP molecular assembly include, besides adenosine triphosphate (ATP) synthase and its subunits [18], adenine nucleotide translocator (ADT). New findings introduced by [19] confirmed the participation of ADT in the mPTP structure. Other mPTP components, namely, voltage-dependent anion channel (VDAC), cyclophilin D (CypD) [20], and a phosphate carrier protein (MPCP) [7], were considered to be involved in the regulation of mPTP activity [8,21,22]. Most proteins that regulate mPTPs directly or indirectly bind to ATP synthase and/or to CypD [23].

Among other regulatory components, members of the Bcl-2 family of proteins and translocator protein (TSPO) were also found, and recently, the involvement of hexokinase-2 (HK-2) and mitochondrial creatine kinase (mtCK) proteins in mPTP regulation was demonstrated [6,12]. The structural role of HK-2 and mtCK consists in stabilization of the contact sites between the inner and outer mitochondrial membranes (IMM and OMM, respectively) through interactions with ADT and VDAC [13]. The cardioprotective effects of mtCK consist in increasing the amount of functional mtCK octamers [24]. Similar protective effects were reported following inhibition of HK-2 dissociation, thereby preventing opening of mPTPs [25]. Understanding the structure–function relationships and regulation of mPTPs is crucial for the development of cardioprotective strategies [26].

At present, mPTP regulation is mostly observed at the level of single changes in single structural or regulatory proteins. So far, however, there has been no consistent information concerning possible interactions between proteins involved in mPTP regulation, which would eventually lead to the protection of the heart from injury.

Functional insight most often requires quantitative comparison between two or more biological states. To fully understand the regulation of protein spectra under both conditions, changes in protein abundances should be analyzed in more detail at the molecular level. While liquid chromatography and mass spectrometry (LC-MS) is not inherently quantitative, this limitation was successfully overcome by the introduction of stable isotopes into the molecules to be identified, or alternatively, by label-free approaches.

In our study, we chose LC-MS-based proteomics to investigate a model of experimental, streptozotocin-induced diabetes mellitus (which is referred to as “experimental D” throughout the text), with the aim of clarifying the cardioprotective function of cardiac mitochondria at the level of mPTP regulation. Shortly after induction, a dramatic increase in blood glucose accompanied by a decrease in blood insulin levels, as well as significant increases in glycohemoglobin, triacylglycerols, and cholesterol, can be detected. Experimental D in the acute stage exhibits all signs of diabetes in the sense that the observed disruption/disorganization of the main metabolic pathways can only be attributed to acute D, as it is not confounded by complications developing due to an ongoing diabetic condition [27,28,29]. For this reason, all adaptation changes developing in the myocardium in the period of acute D are devoid of all the interfering comorbidities incurred by the chronic D condition, and thus can be investigated [30]. The endogenous protective mechanisms present in the acute phase of experimental D were shown to maintain cardiac function at the subcellular level in a similar way to ischemic PC and remote ischemic PC. Therefore, acute-phase experimental D was proven to be a suitable model for the study of endogenous protective mechanisms and can be thought of as a type of metabolic preconditioning (MP) [31,32,33].

The experimental model of D also appears to be well-suited for monitoring possible regulations at the mPTP level. The model described herein involves D-induced damage manifested by an enhanced calcium transient, an increased myocardial energy load, and respiratory chain dysfunction. The progressively accelerated rate of glycolysis in acute-phase D gradually exhausts the supply of oxidized cofactors, resulting in a pseudohypoxia condition when the respiratory chain, despite having enough oxygen, cannot adequately utilize it. The resulting anaerobic glycolysis further exhausts the supply of oxidized cofactors. Hence, the condition of pseudohypoxia produces an excess of lactate, causing the ratio of lactate to pyruvate to elevate, while the ratio of oxidized and reduced form of nicotinamide adenine dinucleotide (NAD+/NADH) falls. On the other hand, this condition also triggers endogenous protective mechanisms to compensate for the damage caused by the noxa, ultimately leading to maintenance of the energy and dynamic balance of the system and to better survival of the acute diabetic myocardium subjected to overload [28,34,35].

Based on the available knowledge, the identification of protein–protein interaction networks and functional interconnection of mPTP-forming and mPTP-regulating proteins may provide interesting insights into the way mPTPs are involved in the cardioprotection process. Furthermore, we aimed to clarify whether cardiac mitochondria could maintain mPTP protein expression at the level of healthy cardiac mitochondria under experimentail D conditions.

## 2. Results

### 2.1. Biochemical Characterization of the Experimental Model

The diabetic status of experimental animals was confirmed on day 8 by statistically significant increases in plasma glucose, blood cholesterol, and triacylglycerols, as well as a significant reduction in blood insulin levels compared to the similarly aged healthy control animals (C) (Table 1). The average starting weight of the rats was 250 ± 12 g. The weight of the diabetic rats was significantly lower by about 20% on average, in comparison with C before the measurements.

### 2.2. Liquid Chromatography and Mass Spectrometry-Based Proteomics

In our proteomic analyses and their quantification, we focused on the proteins that were identified as structural and/or regulatory components of the multiprotein mPTP complex.

#### 2.2.1. Protein Association Network in STRING

First, the scaffold of the association network was constructed by the Search Tool for the Retrieval of Interacting Genes/Proteins (STRING database, version 11.0; https://string-db.org/). STRING is a database of known and predicted protein interactions, including direct (physical) and indirect (functional) associations collected from a number of online databases, as well as from individual high-throughput efforts. All the association evidence is consequently categorized and displayed to allow the user to search for interactions and inspect their patterns. The interaction scores in STRING do not represent the strength or specificity of a given interaction, but rather express an approximate confidence, on a scale of zero to one, of the association being true, given all the available evidence [36]. However, it is important to note that STRING is a predictive database and some of the interactions in the database are only predicted by computational methods (such as Genomic Context) and need to be experimentally verified.

On the basis of available data, the identified mPTP proteins were grouped into functional classes according to the biological processes in which they were known to be involved (Figure 1). The ADT1-VDAC2, ATPB-MPCP, and ATPG-VDAC2 interactions, together with the mutual interactions of ATP synthase subunits, were denoted by the STRING database as significant (i.e., strong).

#### 2.2.2. Hypothesis about the Difference between C and D Conditions with Respect to the Different Expressions (Upregulation and Downregulation of Relevant Proteins)

To semi-quantitatively estimate the protein content within a sample mixture in an LC-MS/MS experiment, the protein abundance index (PAI) was introduced [37] to represent the number of observed peptides divided by the number of theoretically observable tryptic peptides per protein, the latter determined by in silico digestion and subsequent comparison with the MS scan range. It was shown that the number of peptides exhibited a linear relationship with the logarithm of the quantity of injected proteins. Ishihama et al. [38] experimentally proved that the exponentially transformed PAI (emPAI) values were directly proportional to the protein content. Thus, we first utilized the emPAI protocol to estimate the abundance of the proteins identified in the samples. The output data were exported to Excel spreadsheets, and the emPAI values were then calculated and further used to describe and test between-group differences in abundance and expression profiles.

The abundance distribution of all identified mPTP proteins can be seen in Figure 2. The subgroup of low-abundance proteins is centered on the left side, and that of medium-abundance proteins is shifted in the right direction. The remaining highly abundant proteins are located on the right side. The limits of agreement (LoA) were generally defined as the mean difference ± 1.96 SD of difference. If the mean approached zero and the upper and lower limits did not exceed the maximum allowed difference between conditions Δ (i.e., the differences within mean ± 1.96 SD were not biologically important), the two conditions were considered to be in agreement. However, based on the Bland-Altman graph (Figure 2), the abundance of mPTP proteins as a whole was concluded to be significantly different in the D group compared to the healthy C group (*p* = 0.0106). The blue line in Figure 2 indicates mean upregulation (mean—expression of all proteins involved, so the mean value was generated by a mix of downregulated and upregulated proteins; up—because it was above the zero value of no change). The red and brown lines demarcate the estimated precision. The whole region can be seen to lie above zero, which was interpreted as a significant upregulation.

Individually analyzed proteins within the D group, except for ATP5H, ATP5J, and KCRS, did not show significantly altered expressions. The ratios of protein abundances of the two groups (referred to as fold change) are shown in Figure 3 and indicate the relative change of each protein independently, i.e., independent of their absolute abundance. At the level of the two ATP synthase subunits, a slight upregulation was observed, but without statistical significance (Table 2). In the remaining identified proteins involved in the generation and regulation of mPTPs, expression was maintained without significant change at the level of healthy mitochondria.

#### 2.2.3. Differentially Expressed Proteins: Between-Group Comparison

We further proceeded to identify changes in the proteomes of mPTPs in animals subjected to D via the same workflow, taking into account technical replicates and pooling biological replicates to reduce overall subject-to-subject variation. The application of a pooling strategy was based on the assumption of biological averaging, i.e., the assumption that the sources of bias were small and could be neglected. To accomplish the pooling strategy, we analyzed log-transformed emPAI data from two technical replicates measured in two sequential runs on each sample (correlating with molar fraction percentages taken from [38]). The fold change in the expression level of proteins identified under both conditions was calculated by dividing the emPAI value for an individual protein under condition D by the cognate control value C. The log_2_ value of the D/C ratio was calculated and tested against the zero hypothesis of no change, with a margin of equivalence set at 0.585 on the logarithmic scale (log_2_1.5 according to the selected cutoff for biological significance of 1.5-fold). These results are presented in Table 2.

Finally, the absolute fold change was calculated as 2^log^_2_^value^. We demarcated margins for equivalence at a value of 1.5-fold and the cutoff for biological significance at a value of 2.0-fold (Figure 2).

#### 2.2.4. Differentially Expressed Proteins: Within-Group Comparison

After identification of the mPTP structural components present in both D and C conditions, we aimed to estimate or capture the structure and mutual relationships among the identified proteins both within the groups and between the groups. To accomplish this, we performed a so-called co-expression analysis. Differential co-expression methods for analysis of data collected in MS studies can be divided into two categories that serve distinct purposes, i.e., methods that either identify the patterns of correlated proteins, or identify proteins that are differentially expressed under different conditions. Co-expression analysis helps to identify subgroups of proteins that are expressed in a coordinated fashion, i.e., that respond in a similar fashion to the experimentally induced perturbation. Such co-expression is considered as evidence for possible co-regulation and membership to common biological processes [39]. When comparing proteomes between two experimental conditions, it is a natural step to identify differential co-expression to obtain a more informative picture of the dynamic changes in the underlying regulatory networks. Thus, we were interested in the mutual pairwise correlations and the associations between the treatment and each protein identified. In the analysis, every observation X_1,…,18_ was semi-quantitative and expressed in units of emPAI as a measure of relative protein abundance. The response (treatment) was coded as a binary variable, since it represented either the presence or absence of D (i.e., MP) for the corresponding observation.

Since exploratory data analysis relies the most on graphical display, the resulting correlations are presented in Figure 4 in the form of heat-maps, allowing for immediate visual inspection of mutual correlations with significance (indicated by cell borders) and the values of Spearman’s correlation coefficient (represented by colors). The pattern of similarity among correlation structures for these two conditions showed small but distinct deviations from randomness for proteins.

The correlation heat-maps for pooled data and separate within-group correlation coefficients displayed in Figure 4 clearly show changes in the differential co-expression structure of the abundances; subgroups of proteins strongly correlated under one condition but not under the other condition can clearly be distinguished. Differential co-expression may indicate rewiring of transcriptional networks in response to disease or adaptation to different environments [39]. Thus, our results might help to group mPTP components into modules on the basis of their differential co-expression status.

In both groups, the VDAC1–ATP5H relationship was identified as a significant positive correlation. Nevertheless, we identified local differences in the correlation patterns in the C and D groups. The negative correlations of VDAC3 protein with most other proteins identified in the D group did not appear in the C group, while significantly negative correlations of VDAC2 with multiple proteins in the C group did not arise in the D group. The KCRS–ADT1 and VDAC1–ATPB protein relationships followed a positive correlation in the C group, but no relationship was observed between these proteins in the D group. Similarly, significant positive paired correlations of ATP5L with ATPA, ATPO, MPCP, and VDAC1 were present in the D group, but not observed in the C group, or they were negative in some cases. The markedly positive MPCP–ATPO and ATP5J–ADT1 correlations found in the D group were suppressed or reverted in the C group.

#### 2.2.5. Cluster Analysis Results

In proteomic experiments, besides changes in the abundance of a single feature in the MS data, we are often interested in comparing the abundances of a potentially large number of features that are simultaneously detected as part of the experiment. To explore whether biological replicates form “natural” clusters that can be labeled in a meaningful manner, we performed a joining analysis on the standardized emPAI data (features). The resulting dendrogram (cluster analysis tree diagram) began on the bottom, with each case (replicate) in its own cluster (Figure 5A–C). The agglomerative hierarchical method links together cases of increasing dissimilarity (or distance) using Euclidean distance as the distance measure. The method starts by defining each data point as a cluster. Further up in the plot, cases that are close together are combined into a new cluster. In each subsequent step, two existing clusters are merged into a single cluster. Each node in the diagram represents the joining of two or more clusters; the locations of the nodes on the vertical axis represent the distances at which the respective clusters are joined. In this work, the analysis of 18 features successively clustered the cases into two distinct clusters, each with cases more similar within the cluster and dissimilar between clusters (Figure 5A). However, the replicates could not be classified with sufficient reliability, suggesting that part of the features might behave in a similar fashion under both conditions. Generally, group D exhibited lower dissimilarity than group C (Figure 5B vs. Figure 5C), which was consistent with the more “tight” structure of features in comparison with group C (Figure 5D vs. Figure 5E).

Another approach to classification is logistic regression, which belongs to a family named generalized linear models (GLMs) and can be used to predict the class of the biological replicates based on one (simple logistic regression) or multiple predictor variables, i.e., the identified proteins. The outcome (class) is a binary variable with only two possible values, i.e., 0 or 1, or control or diabetic. Logistic regression does not return directly the class of observations, but rather allows the estimation of the probability (P) of class membership. Based on the results from between-group comparisons, we built four simple models to predict the probability of being D-positive based on the abundance of the proteins confirmed as significant in the univariate analysis (Table 2). We found that the classification efficiency of the predictors, which was evaluated as the area under the receiver operating characteristic (ROC) curve, decreased in the order ATP5H (80.6%), KCRS (77.9%), ATP5J (77.8%), and ATPG (69.1%). Their combination did not significantly improve the classification efficiency due to collinearity (e.g., 75.6% for predictors ATP5H and ATPG).

## 3. Discussion

Cardioprotection includes all methods and mechanisms that lead to a reduction in infarct size and thus a reduction in the risk of post-ischemic heart failure. This is a growing research issue, since ischemic heart disease is the leading cause of morbidity and mortality worldwide. A key challenge is the unravelling of cardioprotection complexities, such as the mechanisms leading to prevention of myocardial damage, the cascade of activated cellular signaling pathways, and the cellular and subcellular districts involved, which are tightly interconnected. According to [4], cardioprotection appears to be a highly concerted spatiotemporal program in which different factors with different pathophysiological mechanisms are involved. In this context, cardioprotection can actually be defined as a complex dynamic network of cooperating units, which is characterized by global properties independent of the details of the units in the absence of cooperation [4,40]. This highlights the need to take into account the intertwined actions of the large number of components to better understand the network of cardioprotection.

The aim of the present study was therefore to evaluate whether the cardioprotection conferred by an experimental model of acute D influenced mPTP composition. Since mPTPs form a multienzyme complex, proteomic analysis was considered to be a suitable tool for their characterization. The first references to the existence of mPTPs date back to the 1970s, with several hypotheses about their structure being proposed since. However, the molecular identity of mPTPs remains unclear. Every year, new knowledge about the structure and regulation of this mitochondrial pore appears. Recently, new hypotheses emerged about the nature of mPTPs that form upon conformational change of ATP synthase after Ca^2+^ binding [41].

However, there are inconsistent views on whether the role of ATP synthase can be switched from a key energy-producing enzyme to an energy-dissipating channel, leading to cell death [23,42]. Moreover, a study by Carroll et al. [43] cast doubt on the assumption that ATP synthase is the main structural component, because mPTPs were able to open after the deletion of selected ATP synthase subunits. CypD is a critical regulator of mPTPs and directly binds to the pore constituents, including ATP synthase and ADT, in order to facilitate mPTP opening [44]. CypD–ATP synthase interactions exert functional consequences on enzyme catalysis and can be modulated by cyclosporine A, a direct pharmacological inhibitor of mPTP activity [45]. Other authors also described that increased CypD expression due to increasing age, as well as its interaction with ATP synthase, led to a higher risk of mPTP opening [46]. Various supplements are currently being studied for their ability to influence mPTP regulation. One promising agent, an antioxidant called astaxanthin (AST), appears to have cardiovascular disease-related health benefits as shown in animal studies, as well as potential in the prevention and therapy of other pathological conditions associated with oxidative damage and mitochondrial dysfunction. AST treatment was found to improve the resistance of rat heart mitochondria to Ca^2+^-dependent stress, as well as decrease levels of CypD and increase levels of ADT in mitochondria in a model of experimental myocardial injury induced by isoproterenol [47,48].

The pseudohypoxic myocardium in the investigated experimental model of acute diabetes mellitus with MP features exhibited a slowed-down electron flow in the respiratory chain, resulting in a slowed-down oxidative phosphorylation but without uncoupling oxidation from phosphorylation [28]. These findings point to respiratory-chain damage as a result of the metabolic consequences of diabetes. On the other hand, in the acute phase of diabetes, experimental animals exhibited decreased vulnerability to ischemia [49], reperfusion injury [50,51], and calcium overload [52]. The latter finding was attributed to the increased rigidity of the sarcolemmal membrane due to intensified non-enzymatic glycation of membrane proteins, which decreases the mobility of membrane components and affects the permeability properties of the lipid bilayers. As a result, the myocardium is protected against lethal injury due to severe extracellular calcium overload [52]. Some authors suggested that under experimentally induced load conditions, the diabetic heart adapts [35] and functions even under lower ATP levels, whereas the nondiabetic heart shows lower plasticity and cellular resistance to ischemic injury, suggesting that signaling pathways may have been different in the experimental and C groups [29]. Other findings suggested that increased membrane fluidity along with stimulation of mitochondrial ATP synthase activity could help to improve myocardial energetics during I/R injury. Due to the lack of energy under hypoxic conditions in the acute phase of the disease, the diabetic myocardium activates endogenous protective mechanisms leading to functional mitochondrial remodeling, which involves changes in the chemical and physical properties of the mitochondrial membrane [28,53]. Several studies demonstrated the positive effect of PC on calcium retention capacity and inhibition of mPTP opening. Most authors discussed only selected proteins that form part of the mPTP, with no studies dedicated to components mPTP as a whole and their interactions to consider the effect of PC in conjunction with I/R injury [5,8,25]. The absence of CypD and the presence of mitochondrial bound HK-2 were demonstrated to protect the heart against I/R injury [54]. Several lines of evidence also showed that VDAC played a pivotal role in I/R injury and cardioprotection. Modulation of VDAC by cytosolic proteins was shown to either induce or prevent cell death during I/R injury [55]. HK-2 interactions with VDAC conferred protection in several cardioprotective strategies, such as ischemic PC [56].

Our current work centered on obtaining information that was lacking on the involvement of mPTPs in the processes of endogenous protection. The regulation of the opening and closing of mPTPs is the subject of research by several teams dealing with cardioprotection, but the question of whether regulation is associated with changes in expressions and interactions at the level of individual protein components of the multienzyme mPTP complex, keeping the pores in the closed state, is still unresolved. Using the described experimental model of D, we attempted to determine which changes in the mPTP protein expression could be identified as being associated with the condition that was already proven to trigger adaptation to pathological stimuli. Ultimately, our aim was to find out how these changes contributed to cardioprotective mechanisms that maintain the dynamic equilibrium of the system and ultimately lead to preserving mitochondrial energy metabolism under load conditions represented by acute diabetes mellitus.

Using an LC-MS-based proteomics approach, we identified 18 proteins that were previously described to represent a substantial part of the multienzyme mPTP complex [6,12,13]. In this study, we defined strict inclusion criteria for quantification; therefore, only these proteins were identified in all analyzed samples under the chosen experimental setup (Section 4.7 and Figure 6) and the threshold criteria (Section 4.8). The identified proteins were further quantified with in-built computational tools and the calculated abundances were included in statistical analyses. The analyses determined that ADT1, ATP5H, ATPA, and ATPB had the highest abundances among the identified mPTP proteins, which was consistent with the literature where ADT was reported to be the most abundant protein in the IMM, accounting for up to 10% of the membrane proteins in cardiac mitochondria [57,58]. The analyzed mPTP proteins under the experimental D condition did not exhibit altered expressions relative to the C group. Although ATP5H, ATP5J, and KCRS proteins were found to be significantly upregulated, the deviations were deemed minor. Our findings were in line with the results of unchanged ATPA and ATP5H expressions that were obtained in a similar model by Turko and Murad [59]. Changes in protein expression can trigger signaling pathways leading to cell death, but they can also induce protective responses. In our study, we did not observe changes in the expressions of identified proteins under the condition of acute-phase D. During the progression of diabetes from the acute to the chronic phase, the endogenous protective mechanisms become attenuated and disease-compensatory mechanisms are suppressed. In the chronic phase, the mechanisms of protection at the level of mPTP regulation are further modified, which may ultimately decrease mPTP-regulating protein expression, as evidenced by the proteomic analysis of Baseler et al., which showed a decrease in the expression of four identified ATP synthase subunits as well as other proteins that were part of the respiratory chain during the chronic D phase [60].

Identifying and understanding interaction networks among proteins can facilitate both the elucidation of signaling pathways leading to cardioprotection and the identification of abnormalities in disease. Some protein interactions were also considered by other authors, e.g., Beutner et al. [23] demonstrated that ADT regulated mPTP activity, presumably through interactions with ATP synthase. In addition to ADT, other proteins that were found to regulate mPTPs bound directly or indirectly to ATP synthase and/or CypD, e.g., CypD bound to ATP synthase via the oligomycin sensitivity conferral protein (OSCP) subunit [23].

From the heat-maps and networks, we noted that VDAC proteins were significantly correlated and linked. VDAC isoforms are the most abundant integral membrane proteins of the OMM; these isoforms have spatially and functionally distinct roles and display different expression levels and protein interactive partners [61]. Based on a study performed on isolated rat cardiomyocytes, the existence of a supercomplex composed of ADT, VDAC1, and mtCK was hypothesized [62].

Our analyses yielded a significant positive correlation between VDAC1 and ATP5H in both experimental groups. Several other significant protein correlations were identified, but the correlation pattern differed between the D and C groups, which may indicate that the subtle changes in abundances under pseudohypoxic conditions induced by experimental D were coupled with observed shifts in the intercorrelations.

Our results point to the involvement of mPTP proteins in the endogenous protective processes leading to the preservation of myocardial function under pathological conditions. Proteomic studies with respect to the correlation of mPTP proteins were shown to be one of the most promising options for the advancement of mPTP regulation mechanisms. Subtle changes in mPTP protein expressions, as well as mutual relationships between proteins, may be sufficient to contribute to preserving mitochondrial energy metabolism under the increased energy load represented by experimental D.

## 4. Materials and Methods

### 4.1. Compliance with Ethical Standards

All animal experiments were approved and performed in strict accordance with the rules issued by the State Veterinary Administration of the Slovak Republic, legislation No 377/2012 and with the regulations of the Animal Research and Care Committee ofCentre of Experimental Medicine SAS – Project No. 2237/18-221/3, approved on 21 August 2018.

### 4.2. Experimental Animals

The experiments were performed on 12 male Wistar rats aged 12–15 weeks, weighing 310 ± 20 g (Dobrá voda, Slovak Republic), which were randomly divided into two groups, namely, a control group of healthy rats (C) and a group affected by acute streptozotocin-induced diabetes (D). Animals were housed under natural mode conditions (22 ± 2 °C, 12 h light and 12 h dark) with constant access to drinking water, and were fed a standard pellet diet ad libitum.

### 4.3. Experimentally Induced Acute Streptozotocin Diabetes

Acute diabetes was induced in rats 8 days before the planned experiment by a single administration of streptozotocin (65 mg/kg body weight i.p.) dissolved in 0.1 M citrate buffer (pH 4). The development of the disease during these 8 days, up to the stage of fully developed D (the acute phase of D), was controlled by monitoring glycosuria with Gluko PHAN strips (Pliva-Lachema, Brno, Czech Republic). Blood glucose, cholesterol, and triacylglycerols were determined after excision of the heart from blood collected from the tail vein using the MultiCare diagnostic apparatus (Biochemical system international, Florence, Italy) and appropriate test strips.

### 4.4. Anesthesia

Experimental animals were anesthetized by intraperitoneal administration of thiopental (50–60 mg/kg) applied with heparin (500 IU). They were stabilized for 30 min before the heart was excised from the chest.

### 4.5. Isolation of Mitochondria

Mitochondria were isolated from rat hearts immediately after extraction from the chest. A differential centrifugation method at 4 °C was used for the isolation. Fat parts and aorta were removed from the heart. After adding a small amount of cooled isolation solution (180 mM KCl, 4 mM ethylenediaminetetraacetic acid (EDTA) and 1% serum albumin, pH 7.4), the heart was milled with the GentleMACS Octo Dissociator (2 cycles per 1 min). After addition of the isolation solution to a final volume of 20 mL, the ground heart tissue was homogenized and centrifuged at 1000 *g* for 10 min. The obtained mitochondria-containing supernatant was repeatedly centrifuged at 6200 *g* for 10 min. After the centrifugation, a cold isolation solution (20 mL) without albumin was added to the sediment and finally centrifuged at 6200 *g* for 10 min. The sediment (resulting mitochondrial fraction) was diluted with a small amount of isolation solution without albumin and homogenized [34]. The total protein in the mitochondrial extracts was determined using Bradford’s method [63].

### 4.6. In-Solution Protein Digestion

A volume of mitochondria containing 200 µg of proteins was dried in a centrifugal SpeedVac concentrator. The dried sample was diluted with 8 M urea and 100 mM NH_4_HCO_3_, achieving a urea concentration of 6.4 M in the homogenate. Dithiothreitol (DTT) at a concentration of 30 mM was added as a reducing solution (5 mM DTT in the sample). The solution was then incubated for 30 min at 56 °C in a Thermal Shake lite heater (VWR, Leuven, Belgium). The reduction was followed by alkylation via the addition of 105 mM iodoacetamide (IAA) (15 mM of IAA in the sample), with a subsequent 20 min of incubation in the dark at room temperature. The process of alkylation was stopped by adding 30 mM of DTT to the sample, achieving a concentration of DTT 5 mM. The urea in the sample was diluted to 1.4 M by adding 100 mM NH_4_HCO_3_. The proteins were digested by trypsin (Sigma-Aldrich, St. Louis, MO, USA) at a ratio of 1:20 (a concentration of trypsin 0.2 µg/µL) overnight at 37 °C. The next day, formic acid was added to a concentration of 1% in the sample. The sample was further sonicated and centrifuged at 14,000 *g*.

### 4.7. Analysis by Nano-Liquid Chromatography and Mass Spectrometry (LC-MS)

An Ultimate 3000 RSLC nano System liquid chromatograph (Thermo Fisher Scientific, Germering, Germany) with an Amazon SL, electrospray ionization (ESI) mass spectrometer and a 3D ion trap mass analyzer (Bruker, Bremen, Germany) was used for protein analysis. The mobile phase consisted of component A (0.1% formic acid in 2% acetonitrile (ACN)) and component B (0.1% formic acid in 95% ACN). The loading phase consisted of 0.05% formic acid in 2% ACN. Samples were placed in an autosampler at a temperature of 4 °C and dosed for analysis. The injected amount of the protein was 4 μg. The mobile phase flow rate was 0.4 µL/min with a 265 min gradient in length: 0–35% B for 255 min followed by 10 min at 90% B.

The analyzed samples were separated on a 75 µm × 300 mm column with a 3 µm particle size at 100 Å and 40 °C. The analysis retention time was 265 min and the samples were analyzed in duplicate (technical duplicate). The samples were injected in a partial inject mode. The spectrometer parameters were set as follows: 1450 V capillary voltage, 0.4 Ψ atomizer pressure, 3.0 L/min gas flow rate, 150 °C gas temperature, positive polarity, and scan range 50–2200 m/z at a scan rate of 8100 m/z/sec.

### 4.8. Database Searching and Protein Identification

To define input LC-MS methods and describe analyzed samples, the HyStar program was used. The obtained spectra and chromatograms were processed using Data Analysis software, and correlation analysis was performed by Proteinscape software. The Mascot server 2.5.0. was used as an annotated search library for protein identification. The following parameters were used in the Mascot search: two missed trypsin cleavage sites, carbamidomethyl fixed modification, phosphorylation and oxidation variable modifications, monoisotopic, peptide mass tolerance ±0.6 Da, and MS/MS fragment mass tolerance ±0.6 Da. Peptides were identified and filtered to a 1% false discovery rate (FDR). Each protein identification was required to contain at least one peptide with a Mascot score greater than the Mascot identity score at a significance threshold of *p* < 0.05. The ion score cut-off was set to 15, the peptide rank cut-off was 10, and the minimum peptide length was 5 residues.

To determine functional protein–protein interactions of identified mPTP proteins, we used the STRING database, version 11.0 (https://string-db.org/).

### 4.9. Data Analysis and Interpretation

Direct quantification of proteins in an LC-MS/MS experiment is difficult. However, there is a general correlation between the number of peptides sequenced per protein and the amount of protein present in the mixture. Because larger proteins can give rise to more peptides, a protein abundance index (PAI) was defined [38], which represented the number of peptides identified divided by the number of theoretically observable tryptic peptides. Proteins with two or more identified peptides were further evaluated by the label-free method of the exponentially modified Protein Abundance Index (emPAI), using relative quantitation of the proteins in a mixture based on protein coverage by the peptide matches in a database search result. An excellent correlation was reported between observed emPAI values and independently measured protein copy numbers per cell. For example, protein abundances in the *Escherichia coli* cytosol, as measured by the emPAI approach, correlated well with protein copy numbers per cell measured independently by isotope dilution using spiked *Escherichia coli* BW25113 cells, which contained 40 proteins with known amounts [64].

### 4.10. Statistical Analyses

The retrieved emPAI values were used to create a so-called “fold change” (FC) measure, since in proteomic experiments the true biological measurements are generally viewed as multiplicative in nature, and the use of ratios reflects this assumption. The FC was defined as the ratio of abundances under D/C conditions, averaged across replicates under the D and C conditions. However, the ratios did not provide a natural framework for handling replication. In contrast, analysis of differences in the log-transformed abundances fit into the framework of analysis of variance (ANOVA). On the logarithmic scale (in our case, the logarithm to base 2 of FC was taken), the quantity of interest was the difference Δ between population means μ_D_ and μ_C_, which represented the logarithm of the FC, that is, the logarithm of the ratio of emPAIs between the groups.

### 4.11. Biological Significance Testing

The simplest method for evaluating differences in abundance levels of quantifiable proteins between the D and C conditions is a two-sample *t*-test performed on log-transformed technical replicates, assuming low intersubject variation in the target population of laboratory animals (regarding sex, age, strain, source, housing, care, etc.). Alternatively, the log ratio between the two conditions for each protein can be tested against a biologically meaningful cut-off value for a protein being differentially expressed. For example, if the cut-off value is a two-fold difference, proteins are taken to be differentially expressed if the expression under one condition is over two-fold greater or less than that under the other condition. In our study, we set the cut-off value for statistically significant upregulation (FC = D/C) at 1.5. Its reciprocal 0.66 (or −1.5 if expressed as 1/FC = C/D) was considered as statistically significant downregulation. Changes within the specified range were considered biologically irrelevant.

The statistical model for a design with both biological and technical replicates allowed us to partition the variance of a spectral feature (or log-transformed abundances in a label-free experiment) into separate sources to account for technical and stochastic noise, when the number of animals, sample preparations, and mass spectrum runs were the same in both groups. Therefore, we developed a multistage experimental design with both biological and technical replicates and used a mixed-effect ANOVA with random blocks to analyze the MS data [65].

Cluster analysis was performed to discern cluster patterns of “cases” or features that could be attributed to biological functions. To determine which proteins had the power to differentiate between the conditions, given the observed variance of abundance measurements across all replicates, logistic regression analysis was performed. Prediction models of group membership were developed for a single predictor, as well as for a combination of predictors identified as significant in the between-group comparisons.

Statistical analyses were performed using StatsDirect 3.0.191 software (StatsDirect Ltd., Cheshire, UK) and Statistica 13 software (Dell-StatSoft, Inc. Tulsa, OK, TIBCO Software Inc. USA). All *p*-values were considered statistically significant at a two-tailed *p*-value of <0.05.

## Figures and Tables

**Figure 1 ijms-21-02622-f001:**
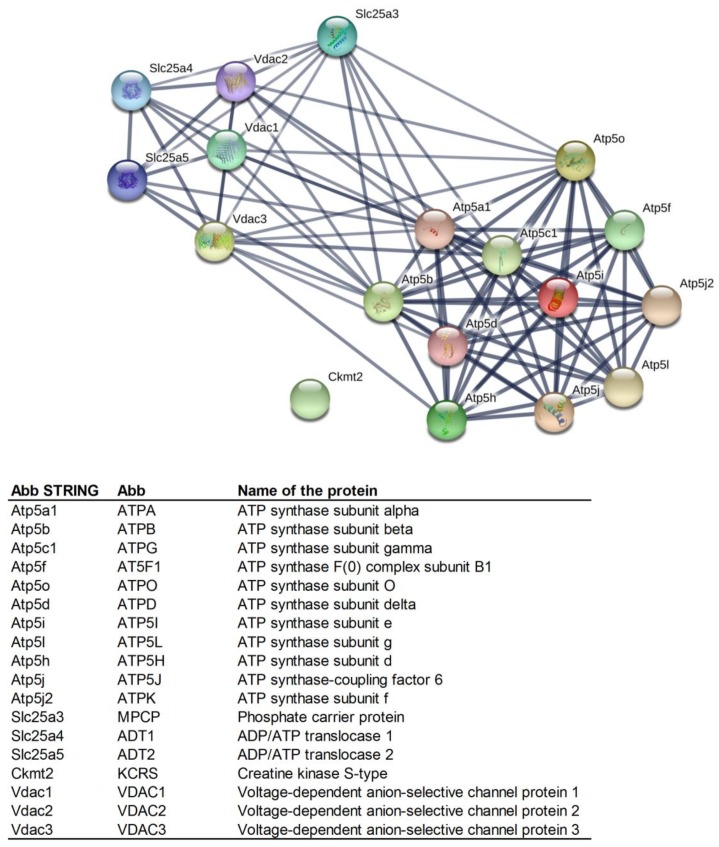
Mutual protein interactions among the proteins identified as being involved in the structure and regulation of mitochondrial permeability transition pores (mPTPs) (created by STRING software) [36]. Minimum required interaction score: high confidence (0.700). Line thickness indicates the strength of data support. Abbreviations of the protein names are in the legend below STRING network.

**Figure 2 ijms-21-02622-f002:**
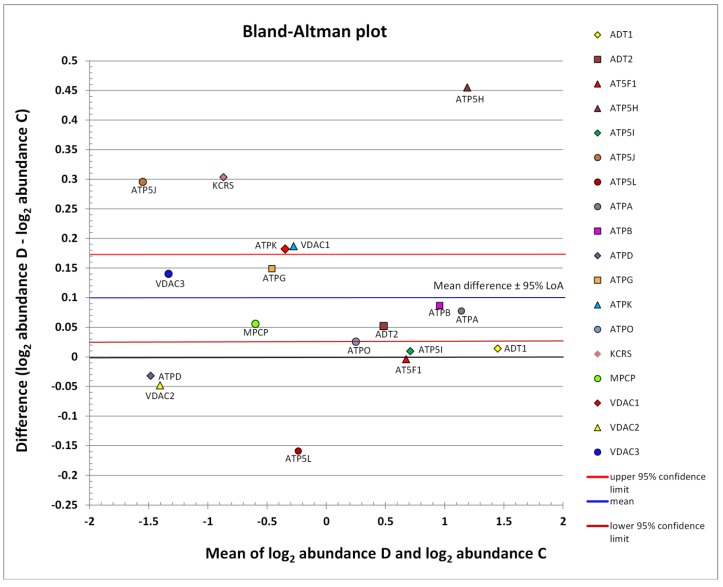
Scatter diagram of the difference between the abundances of a protein in conditions D and C plotted against the average of the two measurements, both on a log scale. Horizontal lines are drawn at the mean difference and at the limits of agreement (95% LoA: 0.0263 to 0.1725). The line of the mean clearly shows an absolute systematic difference of +0.0994 for the investigated mPTP protein components (*n* = 18) in the two conditions (paired *t*-test *p*-value = 0.0106). The most differently abundant protein was ATP5H, with the difference in favor of the D group; this also achieved statistical significance (c.f., Table 2). The other mPTP proteins with the highest average abundance were ADT1, ATPA, and ATPB. Low-abundance proteins with the largest relative change, both in favor of the D condition, were ATP5J and KCRS. Additive effects on the log scale were converted into multiplicative effects on the original scale. The zero value denoted by the black line corresponds to the line of equivalence (c.f., Figure 3).

**Figure 3 ijms-21-02622-f003:**
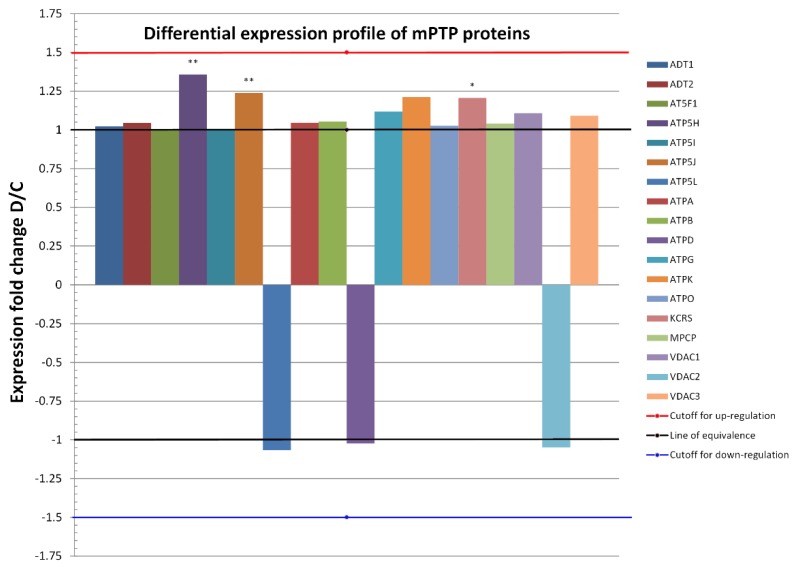
Differential expression profiles of 18 proteins common to both conditions (D and C) in heart mitochondria proteomes. Only proteins that changed in the relative abundance of D to C by at least 1.5-fold (red line), or by −1.5 if expressed as 1/FC = C/D (blue line), were considered as a biologically significant change. Fold changes for the most abundant mPTP proteins, namely, ATPA, ATPB, ATD1, and ATP5H (c.f., Bland–Altman plot), were not significant, except for the latter protein which was significantly upregulated after metabolic preconditioning. On the whole proteome level, 15 upregulated proteins out of 18 proteins (83.3%, exact Clopper-Pearson 95% confidence interval = 58.6% to 96.4%) were identified under the D condition and 3 (16.7%, 95% confidence interval = 3.6% to 41.4%) were identified under the C condition. A binomial test yielded a *p*-value of 0.0075, denoting a statistically significant overall upregulation under the D condition in comparison with controls. * *p* ≤ 0.05, ** *p* ≤ 0.01 compared to C.

**Figure 4 ijms-21-02622-f004:**
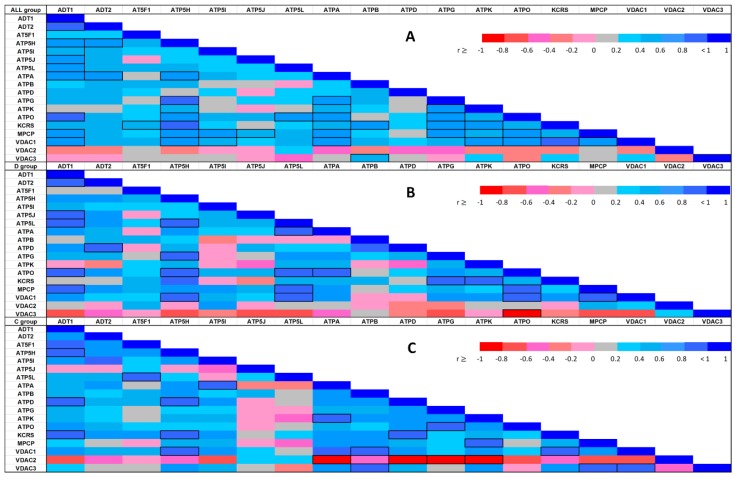
Spearman’s correlation heat-map between each of the identified protein abundances in the whole sample (4**A**: upper heat-map) and stratified by experimental condition (4**B**: middle heat-map for the diabetic condition; 4**C**: lower heat-map for the control group). The color assigned to a point in the heat-map grid indicates the strength of a particular correlation between two protein abundances. The degree of correlation is indicated by red for negative correlations and blue for positive correlations, as depicted in the color key. Statistically significant correlations are marked using cell borders. Note that the upper triangle half of the correlation matrix is symmetrical to the lower triangle half. Thus, there is no need to show the entire matrix.

**Figure 5 ijms-21-02622-f005:**
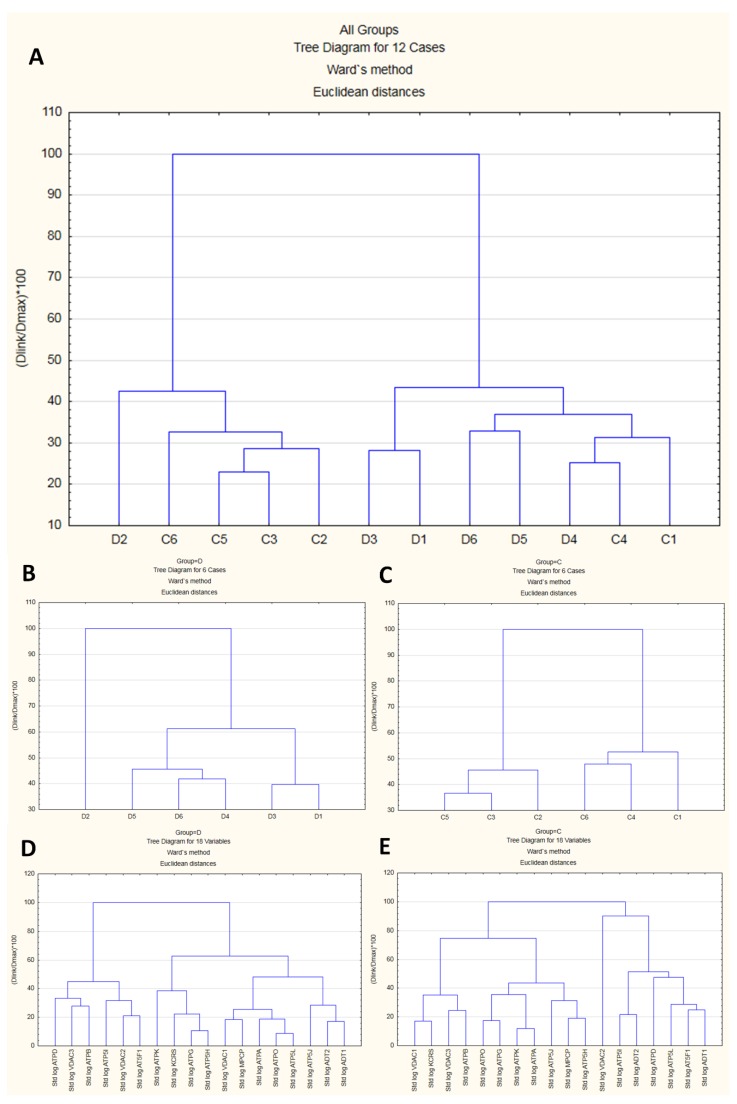
Results of cluster analysis. The upper dendrogram (**A**) shows how sample units (“cases”) plotted on the *x*-axis are combined into “natural” clusters, the height of each branching point corresponding to the normalized distance at which two clusters are joined. The middle left and middle right dendrograms, (**B**) and (**C**), depict the situation in subgroups (**D**) and (**C**), respectively. Similar cases were more strongly associated, i.e., showed smaller distances than dissimilar cases. The lower left and lower right dendrograms, (**D**) and (**E**), depict relationships among 18 mPTP proteins under the D and C conditions, respectively.

**Figure 6 ijms-21-02622-f006:**
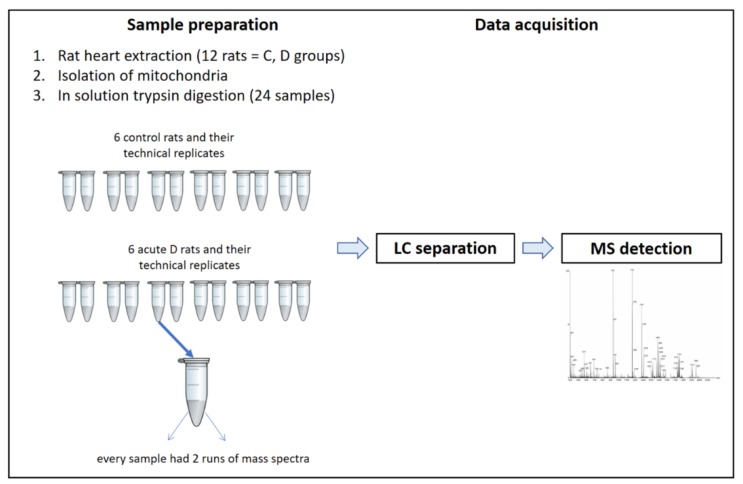
LC-MS workflow.

**Table 1 ijms-21-02622-t001:** Metabolic state of the control and streptozotocin-induced diabetic experimental rats.

Metabolic Parameters	C	D
Glucose (mmol·L^−1^)	6.58 ± 0.48	23.56 ± 0,93 **
Triacylglycerol (g·L^−1^)	1.35 ± 0.17	4.84 ± 0.35 **
Cholesterol (mmol·L^−1^)	1.81 ± 0.17	2.48 ± 0.22 *
Insulin (ng·mL^−1^)	1.08 ± 0.2	0.43 ± 0.12 **

All values are expressed as mean ± standard error of the mean (SEM), *n* = 6 per group. * *p* ≤ 0.05, ** *p* ≤ 0.01 compared to C.

**Table 2 ijms-21-02622-t002:** Analysis of the treatment effects on exponentially transformed protein abundance index (emPAI) values on the logarithmic scale.

Protein	AVE LOG FC D:C	ANTILOG AVE D:C	*p*-Value	Two-Way RM ANOVA		
				Treatment *	Runs *	Interaction *
ADT1	0.0239	1.0167	0.8137	0.8419	0.1511	0.5467
ADT2	0.0592	1.0419	0.686	0.5458	0.0593	0.9187
AT5F1	−0.0012	0.9992	0.9894	0.9567	0.0043	0.3029
ATP5H	0.4415	1.358	0.0021	0.0006	0.0002	0.0111
ATP5I	0.0092	1.0064	0.9582	0.9157	0.0049	0.582
ATP5J	0.3135	1.2427	0.0188	0.0714	0.2404	0.195
ATP5L	−0.0558	0.9621	0.7055	0.3691	0.0763	0.0731
ATPA	0.0746	1.0531	0.4434	0.3389	0.0214	0.1822
ATPB	0.0755	1.0537	0.3352	0.1826	0.0469	0.3286
ATPD	−0.0334	0.9771	0.3519	0.1301	0.455	0.455
ATPG	0.1612	1.1182	0.1221	0.0205	0.0043	0.53
ATPK	0.2865	1.2197	0.2668	0.2875	0.0036	0.8797
ATPO	0.0405	1.0285	0.654	0.7228	0.0001	0.1039
KCRS	0.2803	1.2145	0.0699	0.0002	<0.0001	0.0195
MPCP	0.0597	1.0423	0.1186	0.3071	0.6687	0.586
VDAC1	0.1558	1.114	0.3914	0.0497	0.005	0.1429
VDAC2	−0.0692	0.9531	0.4733	0.6699	0.2726	0.1745
VDAC3	0.1216	1.088	0.1376	0.2279	0.2051	0.1149

* *p*-value from testing the hypothesis of no difference in log emPAI values for the indicated sources of variation. FC: fold change; AVE: average; P: probability; RM: repeated measures.

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
