# Peer review of "mPTP Proteins Regulated by Streptozotocin-Induced Diabetes Mellitus Are Effectively Involved in the Processes of Maintaining Myocardial Metabolic Adaptation"

_ijms, 2020, doi:10.3390/ijms21072622_

Round 1

Reviewer 1 Report

The paper of Andelova et al is a description proteins of  mitochondrial permeability transition pore that regulated by streptozotocin-induced diabetes mellitus are effectively involved in the processes of maintaining myocardial metabolic adaptation. The paper is well presented and well written. However, there are some comments on submitting the article.

1) Panels in Figures 4 (A-C) and 5 (A-E) are not indicated, although they are indicated in the figure legends and described in the results.

2) Cyclophilin D is the structural component of mPTP. This protein is associated with Complex III and ATP synthase (Giorgio, V.; Bisetto, E.; Soriano, M.E.; Dabbeni-Sala, F.; Basso, E.; Petronilli, V.; Forte, M.A.; Bernardi, P.; Lippe, G. Cyclophilin d modulates mitochondrial f0f1-atp synthase by interacting with the lateral stalk of the complex. J. Biol. Chem. 2009, 284, 33982–33988; Laker, R.C.; Taddeo, E.P.; Akhtar, Y.N.; Zhang, M.; Hoehn, K.L.; Yan, Z. The mitochondrial permeability transition pore regulator cyclophilin d exhibits tissue-specific control of metabolic homeostasis. PLoS One 2016, 11, e0167910; Olga Krestinina, Yulia Baburina, Roman Krestinin, Irina Odinokova, Irina Fadeeva, Linda Sotnikova. Astaxanthin Prevents Mitochondrial Impairment Induced by Isoproterenol in Isolated Rat Heart Mitochondria // Antioxidants. 2020. 9(3). 262; Yulia Baburina, Roman Krestinin, Irina Odinokova , Linda Sotnikova, Alexey Kruglov and Olga Krestinina Astaxanthin Inhibits Mitochondrial Permeability Transition Pore Opening in Rat Heart Mitochondria // Antioxidants. 2019. 8 (12). 576).

Authors mention Cyclophilin D in “Introduction” but do not consider it in their research. Why? It is possible, the authors will perfom such research in the future. It is difficult to repeat the entire study for this protein, however, it is an important protein in the regulation of mPTP and it should be discussed in "Discussion". 

Author Response

Dear Editors,

we would like to thank all Reviewers for their opinions, critical comments, suggestions, as well as for time you all spent on our manuscript.

We have revised the original text of the manuscript according to the reviewers' comments.

We believe that the corrections added to the manuscript and the clarifications in the accompanying text have met the reviewer's requirements and clarified the manuscript. We hope that the revised manuscript will be suitable for publication. Thank you and we wish good health and a lot of strength to everyone.

Please find enclosed the revised version of manuscript entitled: “mPTP proteins regulated by streptozotocin-induced diabetes mellitus are effectively involved in the processes of maintaining myocardial metabolic adaptation”.  Changes and the added text are indicated by highlighting using Track Changes function.

Sincerely yours,

Ing. Miroslav Ferko, PhD.

Center of Experimental Medicine Slovak Academy of Sciences

Institute for Heart Research, Bratislava, Slovak Republic

We thank all Reviewers for their opinion and valuable comments which helped us to improve reasoning and presentation of our findings.

Manuscript ID: ijms-764256, Rev. 1

We thank to the Reviewer for her/his comments and suggestions and hope that she/he will consider our additions useful.

1) Panels in Figures 4 (A-C) and 5 (A-E) are not indicated, although they are indicated in the figure legends and described in the results.

Thank you, we apologize for overlooking it. We have indicated the panels in Figures 4 (A-C) and 5 (A-E).

2) Authors mention Cyclophilin D in “Introduction” but do not consider it in their research. Why? It is possible, the authors will perform such research in the future. It is difficult to repeat the entire study for this protein, however, it is an important protein in the regulation of mPTP and it should be discussed in "Discussion".

Thank you very much for the inspiring question. As pointed out by the reviewer, CypD is one of the most important components of mPTP. We agree that it would be appropriate to include this protein in the analysis of results, however, in this study, we defined strict inclusion criteria for quantification, therefore, only those proteins that were identified in all analyzed samples and their replicates under chosen experimental setup (section 4.7 and Figure 6) and meet the threshold criteria (section 4.8), were also quantified to provide an assessment of possible between-group differences. For this reason, we could not include CypD in the quantification in this study. However, as the reviewer mentioned, it is our goal for the future – likely with some modification of experimental procedures and setup, since this protein needs to be analyzed in a different way, via MRM transitions (more precise than spectral counting), which will allow us to obtain information about the change of the protein through the peak area determination. This type of analysis was not possible in the first series of measurements in terms of the amount of mitochondria required for analysis. For this yield limitation, we could not perform individual protein analyses via MRM transitions in addition to the MS-MS analyses.

The essentials of state-of-the-art knowledge about regulatory component mPTP - CypD have been incorporated into the Discussion part. We have also elaborated suggested references and highlighted them in the text (line 338).

Reviewer 2 Report

I read the manuscript with much expectations and enthusiasm. However, I am slightly disappointed that I feel the overall manuscript is a story half-told.

I would love to see how these changes in mPTP proteins relate to functional outcomes, for example how different are they under ischaemic conditions or when compared to other preconditioning. Or, how do these "protective" changes differ in long-term?

There are also some minor styling error, e.g. line 313.

Author Response

Dear Editors,

we would like to thank all Reviewers for their opinions, critical comments, suggestions, as well as for time you all spent on our manuscript.

We have revised the original text of the manuscript according to the reviewers' comments.

We believe that the corrections added to the manuscript and the clarifications in the accompanying text have met the reviewer's requirements and clarified the manuscript. We hope that the revised manuscript will be suitable for publication. Thank you and we wish good health and a lot of strength to everyone.

Please find enclosed the revised version of manuscript entitled: “mPTP proteins regulated by streptozotocin-induced diabetes mellitus are effectively involved in the processes of maintaining myocardial metabolic adaptation”.  Changes and the added text are indicated by highlighting using Track Changes function.

Sincerely yours,

Ing. Miroslav Ferko, PhD.

Center of Experimental Medicine Slovak Academy of Sciences

Institute for Heart Research, Bratislava, Slovak Republic

We thank all Reviewers for their opinion and valuable comments which helped us to improve reasoning and presentation of our findings.

Manuscript ID: ijms-764256, Rev. 2

We thank to the Reviewer for her/his comments and suggestions and hope that she/he will consider our additions useful.

1) I would love to see how these changes in mPTP proteins relate to functional outcomes, for example how different are they under ischaemic conditions or when compared to other preconditioning. Or, how do these "protective" changes differ in long-term?

Most authors are focused on studying the influence of various substances or stimuli on the modulation of mPTP opening, or on some specific mPTP-forming proteins. Our research of the literature available through remote access offered by academic libraries did not retrieve studies which had dealt with individual mPTP components as a whole, which would, in addition, consider experimental condition and methods under question (effect of preconditioning (PC) in conjunction with I/R injury using a complex LC-MS/MS analysis). The overall positive effect of PC associated with inhibition of mPTP opening has long been known. The model of experimental STZ-diabetes is known to have similar signaling pathway pattern to PC and therefore we chose it in our study in order to monitor and elucidate the adaptation mechanisms at the level of individual mPTP proteins and their interactions. It is not yet clear how individual mPTPs behave due to ischemic damage or some other form of PC. However, we are planning to study the behavior of individual mPTP proteins and their interactions in the I/R and remote ischemic PC models, too. How individual mPTP proteins would change in the advanced stages of diabetes is also unclear, but it is known that the expression of ATP synthase subunits decreases in chronic stages of diabetes, as conferred in the Discussion.

We have also added the results of studies in which the authors have monitored changes at the level of selected mPTP proteins in relation to PC and I/R injury, because complex studies of mPTP proteins as a whole do not yet provide this knowledge. New informations and references are elaborated in the Discussion, highlighted in the text (line 370, 393).

2) There are also some minor styling error…

Thank you for the notice, we have go through and corrected some minor styling errors and inadvertences in the manuscript.

 In the original Figure 3 we have highlighted the threshold lines which were missed in the processing the image to the 300 dpi resolution, so we are sending this Figure again.